# Detection of Various *Streptococcus* spp. and Their Antimicrobial Resistance Patterns in Clinical Specimens from Austrian Swine Stocks

**DOI:** 10.3390/antibiotics9120893

**Published:** 2020-12-11

**Authors:** René Renzhammer, Igor Loncaric, Marisa Ladstätter, Beate Pinior, Franz-Ferdinand Roch, Joachim Spergser, Andrea Ladinig, Christine Unterweger

**Affiliations:** 1University Clinic for Swine, Department for Farm Animals and Veterinary Public Health, University of Veterinary Medicine, 1210 Vienna, Austria; schweineklinik@vetmeduni.ac.at (M.L.); andrea.ladinig@vetmeduni.ac.at (A.L.); christine.unterweger@vetmeduni.ac.at (C.U.); 2Department for Pathobiology, Institute of Microbiology, University of Veterinary Medicine, 1210 Vienna, Austria; loncarici@staff.vetmeduni.ac.at (I.L.); i102us01@staff.vetmeduni.ac.at (J.S.); 3Unit of Veterinary Public Health and Epidemiology, Department for Farm Animals and Veterinary Public Health, Institute of Food Safety, Food Technology and Veterinary Public Health, University of Veterinary Medicine, 1210 Vienna, Austria; Beate.Pinior@vetmeduni.ac.at (B.P.); Franz-Ferdinand.Roch@vetmeduni.ac.at (F.-F.R.)

**Keywords:** swine, *Streptococcus*, SDSE, *S. hyovaginalis*, *S. thoraltensis*, resistance, tetracyclines

## Abstract

Knowledge of pathogenic potential, frequency and antimicrobial resistance patterns of porcine *Streptococcus* (*S*.) spp. other than *S. suis* is scarce. Between 2016 and 2020, altogether 553 *S*. spp. isolates were recovered from clinical specimens taken from Austrian swine stocks and submitted for routine microbiological examination. Antimicrobial susceptibility testing towards eight antimicrobial substances was performed using disk diffusion test. All isolates from skin lesions belonged to the species *S. dysgalactiae* subspecies *equisimilis* (SDSE). *S. hyovaginalis* was mainly isolated from the upper respiratory tract (15/19) and *S. thoraltensis* from the genitourinary tract (11/15). The majority of *S. suis* isolates were resistant to tetracycline (66%), clindamycin (62%) and erythromycin (58%). *S. suis* isolates from the joints had the highest resistance rates. *S. suis* and SDSE isolates resistant to tetracycline were more likely to be resistant to erythromycin and clindamycin (*p* < 0.01). Results show that different species of *Streptococcus* tend to occur in specific body sites. Nevertheless, a statement whether these species are colonizers or potential pathogens cannot be given so far. High resistance rates of *S. suis* towards tetracyclines and erythromycin and high recovery rates of *S. suis* from lung tissue should be considered when treating pigs with respiratory diseases.

## 1. Introduction

The genus *Streptococcus* (*S*.) is composed of Gram-positive, catalase negative, facultative anaerobes [1]. *S. suis* plays a major role in animal and human medicine due to its zoonotic potential [2]. In general, infections of weaned piglets with strains of *S. suis* often result in pneumonia, septicemia, meningitis, arthritis or serositis leading to increased mortality rates and high economic losses [1,2]. Besides *S. suis* also other *Streptococcus* species are commonly recovered from organs of clinically diseased pigs [3,4,5,6]. However, since it is still unclear, if these species can cause clinical symptoms interpretation remains difficult.

In this study, we describe the detection of *Streptococcus* (*S.*) *dysgalactiae* subspecies *equisimilis* (SDSE), *S. hyovaginalis*, *S. thoraltensis*, *S. porcinus*, *S. orisratti* and *S. alactolyticus.* SDSE is part of the physiological flora of gilts and sows and can be frequently recovered from vaginal secretions and colostrum [7], which are described to be the most common sources of infections of piglets [8]. SDSE is able to cause septicemia after entering the blood stream via tonsils or skin lesions of piglets and can further lead to arthritis, meningitis or endocarditis [4]. *S. hyovaginalis* and *S. thoraltensis* have mainly been recovered from the reproductive tract of clinically healthy sows so far [3]. SDSE and *S. thoraltensis* both have been isolated from organs of diseased humans working in the meat industry. Therefore, the detection of these streptococcal species in clinically diseased pigs is of major interest for human and animal health [9,10,11]. Since there are only few reports on the isolation of *S. porcinus* [12,13], *S. orisratti* [6,14] or *S. alactolyticus* [3] from pigs, their pathogenic potential remains unknown.

On the other hand, antimicrobial resistance of *S. suis* is emerging which brings up concerns primarily due to their zoonotic potential [15], but also regarding animal welfare and the appropriate treatment of diseased pigs. After the first report of penicillin-resistant *S. suis* isolates in 1980 [16], several investigations confirmed the occurrence of *S. suis* isolates with resistances to penicillins [17,18,19]. Many studies reported high resistance rates of *S. suis* to tetracycline, clindamycin and erythromycin, whereas resistance rates to beta-lactam antibiotics and quinolones were low in most investigations [17,18,19]. In contrast, resistances of SDSE have mainly been described for tetracyclines and macrolides but also for quinolones [10,20]. However, there are few data records available about the resistance rates of *S. hyovaginalis*, *S. thoraltensis*, *S. porcinus*, *S. orisratti* or *S. alactolyticus* towards commonly applied antibiotics. Therefore, knowledge about susceptibility patterns of these bacteria is needed, especially due to the potential of some species to cause diseases in pigs and humans.

The aim of this study was to provide an overview of the antimicrobial resistance patterns and the distribution of various species of *S.* spp. in different organs, isolated from clinically diseased pigs, in order to determine the potential pathogenic impact of *S.* spp. other than *S. suis*.

## 2. Results

### 2.1. Distribution of Different Species of Streptococcus 

Out of all 553 *S.* spp. isolates, *S. suis* was the most frequent species (*n* = 297, 54%), followed by SDSE (*n* = 158, 29%) (Table 1). At least one isolate belonging to the genus *Streptococcus* was recovered from 52% (211/428) of all investigated lung samples and from 45% (57/128) of all examined swab samples taken from the upper respiratory tract. Furthermore, *S.* spp. were isolated from 34% (83/244) of all examined specimens of the genitourinary tract of sows with reproductive failure and approximately from one quarter of swabs taken from serosal surfaces (55/217), joints (26/109) and skin lesions (13/49) (Table 1).

SDSE isolates were significantly more often recovered from specimens taken from the genitourinary tract than *S. suis* (Standardized residuals 3.47) and less frequently from samples of the upper respiratory tract (Standardized residuals −2.50) in a chi squared frequency distribution. In total, SDSE was recovered from 28 organ pools of aborted fetuses, seven cervical swabs and six vaginal swabs.

*S. hyovaginalis* was mainly isolated from the upper respiratory tract (15/20) but never from the genitourinary tract or abortion material, whereas eleven out of all 15 *S. thoraltensis* isolates were recovered from the genitourinary tract of sows with reproductive disorders (73%) (Table 1). Two *S. hyovaginalis* isolates were recovered from meningeal swabs derived from weaned piglets with central nervous disorders. *S. porcinus* was almost exclusively recovered from the respiratory tract (11/12) of weaned piglets (Figure 1). Compared to all other species, *S. alactolyticus* was isolated disproportionately more frequently in specimens from the serosal surfaces than from other tissues. All *S. alactolyticus-positive* samples from serosal surfaces originated from weaned piglets expressing symptoms such as wasting and increased mortality rates (Figure 1).

Although all species were recovered from the respiratory tract, their occurrence in the upper and lower respiratory tract varied. *S. hyovaginalis, S. thoraltensis* and *S. porcinus* were more commonly found in the upper respiratory tract and *S. suis*, SDSE and *S. alactolyticus* were more frequently recovered from the lower respiratory tract (Figure 2). With the exception of *S. thoraltensis,* all species were isolated from lung samples from clinically diseased pigs. The majority of *Streptococcus* species deriving from the upper respiratory tract were *S. suis* (50%) and *S. hyovaginalis* (27%), while SDSE was the most frequently detected *Streptococcus* species in the genitourinary tract of sows expressing reproductive disorders or abortion material (49%), followed by *S. suis* (24%) and *S. thoraltensis* (13%). Out of all eleven *S. thoraltensis* isolates that were recovered from sows with reproductive problems eight were recovered from organs of aborted fetuses and three from cervical swabs.

*S. suis* was the most frequently isolated *Streptococcus* species in specimens of the central nervous system (84%). The distribution of species in the joints and serosal surfaces was similar with *S. suis* as the most frequently isolated species (42%, 50%), followed by SDSE (39%, 32%) and *S. alactolyticus* (8%, 10%), whereas *S. hyovaginalis*, *S. thoraltensis* and *S. porcinus* were neither detected in joints nor on serosal surfaces. SDSE was the only *Streptococcus* species that was isolated from skin lesions (Figure 2).

### 2.2. Antimicrobial Resistance Patterns

Two *S. suis* isolates were resistant to penicillin, whereas no isolate was resistant to ceftiofur (Table 2). Out of all *S. alactolyticus* isolates that were resistant to penicillin (29%), one was also resistant to ceftiofur. Resistances to beta-lactam antibiotics were not observed for any other species (Table 2). Resistances to levofloxacin were observed occasionally for *S. suis* (4/297) and SDSE (1/158) (Table 2).

All *S. suis* isolates were susceptible to linezolid and vancomycin. The majority of *S. suis* isolates was resistant to tetracycline (66%), clindamycin (62%) and erythromycin (58%) (Table 2). Resistance rates of all other six investigated *Streptococcus* species towards these three antibiotics were also higher than to other substances, although resistance rates of SDSE, *S. porcinus* and *S. alactolyticus* to erythromycin were below 50%. Generally, SDSE had lower resistance rates than *S. suis* to all tested antimicrobial substances. In total, 95% of all *S. hyovaginalis* isolates were resistant to tetracycline, erythromycin and clindamycin. *S. porcinus* isolates had the lowest resistance rates to clindamycin (25%) and erythromycin (17%). All detected *S. orisratti* isolates were resistant to tetracycline. In total, 29% of all *S. alactolyticus* isolates were resistant to penicillin. *S. suis* and SDSE isolates resistant to tetracycline were also more likely to be resistant to erythromycin and clindamycin (*p* < 0.01). Resistance rates of SDSE (51/143) towards erythromycin were significantly lower than those of *S. suis* (172/297) (*p* = 0.01).

*S. suis* isolates from the upper respiratory tract and joints had generally higher resistance rates to erythromycin and clindamycin compared to isolates from the lower respiratory tract, the central nervous system or serosal surfaces. Isolates from the genitourinary tract had the second highest resistance rate to tetracyclines (78%) but the lowest number of isolates resistant to erythromycin and clindamycin (Table 3).

Since there are no criteria available for the interpretation of resistances against trimethoprim-sulfamethoxazole, data about these substances are not presented. However, since trimethoprim-sulfamethoxazole is commonly applied to treat swine, resistance rates to these substances were also evaluated in routine diagnostics. In total, 75% of all isolates were resistant to trimethoprim-sulfamethoxazole.

## 3. Discussion

This is a retrospective study providing data collected in the course of routine diagnostic procedures over 4 ½ years. Since all samples were taken from clinically affected pigs our results cannot reflect the current prevalence of *Streptococcus* spp. in Austrian swine stocks but rather provide an overview of the occurrence of certain *Streptococcus* spp. in diseased pigs. Although we were able to detect those species in diseased animals, results cannot demonstrate any causality between clinical symptoms and the recovery of the respective isolates. Furthermore, we do not have sufficient information about other primary infectious agents or previous antimicrobial treatments, which could have influenced our results impactfully. Other diagnostic methods to detect pathogens such as viruses or *Mycoplasma* were only performed if requested by the veterinarian who has submitted the specimens. Despite all these limitations of the study design, it is nevertheless worthwhile to publish these data, especially since there is hardly any information on *Streptococcus* species other than *S. suis* available in diseased swine.

Although *S. suis* usually occurs more often in the upper than in the lower respiratory tract [21], *S*. *suis* was detected more frequently in lung samples than in nasal and tonsillar swabs. Therefore, it is impossible to state whether these *S. suis* isolates have been in the lung tissue prior to sampling or if isolation was just the result of contamination afterwards. Secondly, almost all lung samples were taken from clinically diseased pigs, which often underwent infections with other pathogens such as *Mycoplasma hyopneumoniae*, Influenza-A Virus or Porcine Reproductive and Respiratory Syndrome Virus (PRRSV) predisposing lung tissue for the invasion of other pathogens such as *S. suis*. Since various *Streptococcus* species are described to be colonizers of the upper respiratory tract, the number of isolates recovered from the upper respiratory tract was lower than expected. A clear statement, whether these observations are a result of prior antimicrobial treatments or not, cannot be given.

SDSE was the only species that was recovered from all investigated body sites (Table 1). Its ability to cause septicemia followed by its spread to joints and the CNS has been described previously [22]. The detection of SDSE in cervical swabs could at least indicate its presence in the uterus and its potential involvement in reproductive disorders as well, since the risk of contamination using swabs for mares is rather low. However, if SDSE can cause reproductive disorders such as abortions on its own, needs to be further investigated. SDSE has also been isolated from porcine skin lesions in a previous study [23]. Abscesses and other skin lesions are often associated with infections with *Streptococcus* spp. However, there is hardly any information about the presence of SDSE on skin lesions. Remarkably, SDSE was the only *S.* spp. recovered from these samples.

In a study conducted by Brazilian colleagues, *S. thoraltensis* has been recovered from specimens of the genitourinary tract of sows with reproductive disorders [5]. Since *S. thoraltensis* was isolated several times from cervical swabs, we assume the presence of *S. thoraltensis* also in the cervical mucosa of sows. A case report about an acute chorioamnionitis of a healthy woman due to an infection with *S. thoraltensis* [11] emphasizes its potential to cause reproductive disorders in various hosts.

In contrast to *S. thoraltensis,* we could not recover *S. hyovaginalis* from any specimen of the genitourinary tract. However, in previous reports *S. hyovaginalis* was exclusively isolated from specimens of the genitourinary tract of sows but never from the respiratory tract [3,5]. Nevertheless, in those studies *S. hyovaginalis* was solely recovered from vaginal discharge, whereas most of the specimens taken from the genitourinary tract in our study were cervical swabs and organs of aborted fetuses. There are no reports about the detection of *S. hyovaginalis* in the uterus or organs of fetuses. Therefore, *S. hyovaginalis* might be a colonizer of the nasal and vaginal mucosa but might not be able to enter the lower respiratory tract or uterus under normal circumstances. Both weaned piglets from which we were able to recover *S. hyovaginalis* from the lung tissue were also infected with PRRSV. We cannot state whether the isolation of *S. hyovaginalis* from one meningeal swab was the result of a contamination or not. However, since no other bacteria could be isolated from the same swab, contamination seems rather unlikely.

Despite reports on the isolation of *S. porcinus* in tonsils of 20% of monitored slaughtered pigs in Canada [6] and 38% of carcasses in Russia [24], *S. porcinus* was recovered in only three out of 128 investigated specimens from the upper respiratory tract. Whether *S. porcinus* is a colonizer of the lower respiratory tissue as well or the detection of *S. porcinus* in the lungs is a result of its invasion after the induction of lung lesions caused by infections with primary pathogens, is still debated. *S. porcinus* has already been associated with lymphadenitis and cervical abscesses [14]. Before the distinction between *S. pseudoporcinus* from *S. porcinus*, *S. porcinus* was considered as a pathogen with zoonotic potential [25]. However, since the awareness that all zoonotic isolates belong to the species *S. pseudoporcinus*, the global interest on *S. porcinus* is declining rapidly. Thus, hardly any investigations on *S. porcinus* have been published recently. In contrast to previous reports [26], we could not recover *S. porcinus* from the CNS, urogenital tract or joints.

In total, only ten *S. orisratti* isolates were recovered mainly from the respiratory system. However, one isolate was recovered from synovial fluid. Due to its very similar characteristics to *S. suis*, it might also possess the ability to cause septicemia leading to lesions in joints and other organs [12]. Since *S. orisratti* has mainly been recovered from the teeth of rats and *Rattus norvegicus* is generally common in Austrian swine stables, contamination of *S. orisratti* due to gnawing on carcasses of dead piglets cannot be disregarded.

Little is known about the impact of *S. alactoylticus* and its potential involvement in the pathogenesis of diseases in pigs. In a previous Brazilian study, *S. alactolyticus* was also recovered from the respiratory and genitourinary tract [5]. However, we did not recover *S. alactolyticus* from any specimen of the upper respiratory tract. *S. alactolyticus* and its heterotypic synonym *S. intestinalis* have been recovered from various porcine specimens including intestines, feces, lungs and vaginal discharge [5]. Furthermore, it is also known to colonize the intestines of other animals including pigeons and chickens and was reported to be the most common lactic acid bacteria (LAB) in the intestines of canines [27]. Since we are aware of the presence of *S. alactolyticus* in the feces and *E. coli* was isolated from four out of six serosal swabs from which *S. alactolyticus* could be detected, recovery of the strains might just be the result of contamination. However, in the two remaining swabs only *S. alactolyticus* was isolated. This goes along with the isolation of *S. alactolyticus* from the synovia or synovial membranes. In fact, *S. suis,* SDSE, *S. orisratti* and *S. alactolyticus* were the only four species that were recovered from joints or serosal surfaces. The potential of *S. suis* and SDSE to cause septicemia is known. However, *S. alactolyticus* might be able to spread through the body of infected pigs as well. Generally, isolates belonging to these species are often recovered from organs of clinically diseased pigs in the absence of any other infectious agent potentially causing the clinical conditions. However, it remains speculative if affected animals should receive antimicrobial treatment proposed by the determined resistance patterns of the investigated isolate or not.

In contrast, *S. hyovaginalis* was mainly recovered from the upper respiratory tract and *S. thoraltensis* from the genitourinary tract indicating that both species are colonizers of the respective organs, without causing septicemia. In general, more invasive species such as *S. suis,* SDSE and *S. alactolyticus* were recovered more frequently from the lower respiratory tract, whereas *S. porcinus*, *S. hyovaginalis* and *S. thoraltensis* were recovered more frequently from the upper respiratory tract (Figure 2). A causal association between clinical symptoms and the recovery of different *Streptococcus species* could not be demonstrated and can only be proven by an experimental setting.

For antimicrobial susceptibility testing agar disk diffusion test was applied. For penicillin clinical break points of beta hemolytic streptococci were used. We are aware that the results obtained in this way cannot predict the susceptibility of viridans streptococci. Therefore, results of non beta-hemolytic *S.* spp. should be interpreted with caution. Furthermore, interpretative criteria for ceftiofur were only available for isolates from the respiratory tract. However, veterinarians have to work with these results in the field. Thus, more knowledge about interpretative criteria for *S.* spp. is urgently needed. In general, low numbers of isolates being resistant to beta-lactam antibiotics but high numbers of porcine clinical *S. suis* isolates being resistant to tetracyclines, clindamycin and erythromycin have been reported previously [17,18,28,29]. In addition, resistance rates for quinolones varied between 1% and 44% amongst those studies [17,18,19,29]. Although only two *S. suis* isolates showed an in vitro resistance to penicillin, penicillin resistant *Streptococcus* species seem to be on the rise [17,18]. However, treatment with penicillin still led to good responses, even if isolated strains had shown an in vitro resistance in disk diffusion test [30]. Since resistance rates towards penicillin were generally low, there is no need to choose a critical antibiotic such as ceftiofur for the treatment of streptococcosis. Strains recovered from tonsils are described to be more frequently multi resistant than strains from other body sites [29], which was also true in our observations. High numbers of resistant isolates from the joints could be the result of long-term treatment. There might be a selection towards multi resistant *S. suis* strains in the joints, since not all antimicrobial metabolites could cross the blood-synovial barrier appropriately and arthritis is mainly an accompanying symptom of chronic infections with *S. suis*. Generally, tetracyclines and macrolides are widely applied to treat respiratory diseases in pigs [31]. However, *S. suis* was recovered in approximately 50% of all lungs from weaned piglets and fattening pigs with respiratory diseases and resistance rates towards tetracyclines and erythromycin were high in our investigations. Therefore, susceptibility testing or a reconsideration of choice of antimicrobial substance of these isolates is highly recommended. Since trimethoprim-sulfamethoxazole is also frequently applied to treat swine stocks [31] and resistance rates evaluated for the routine diagnostics seemed to be quite high, there is definitely a need for valid interpretative criteria for these substances.

Moderate–high resistance levels of SDSE to tetracyclines [10] and macrolides [32] were also reported before.

Knowledge about antimicrobial resistances of other discussed species is scarce. However, in previous investigations in Brasilia, all 13 *S. hyovaginalis* isolates had high MIC values for tetracyclines, macrolides and clindamycin and low MIC values for beta-lactam antibiotics [5]. This goes in line with our findings, since resistance rates of *S. hyovaginalis* to erythromycin, tetracycline and clindamycin (each 95%) were higher than the resistance rates of *S. suis* or SDSE to the respective antimicrobial substances.

Interestingly, over 90% of all isolates were resistant to trimepthoprim-sulfamethoxazole. High detection rates of *S. thoraltensis* and case reports of infections in humans could emphasize its impact on the pathogenesis of reproductive disorders. In general, diagnostics of reproductive disorders in sows are difficult, especially regarding *Leptospira interrogans.* Although tetracyclines are most frequently applied to sows with the suspicion of infections with *Leptospira interrogans* or *Chlamydia* spp. [33] they might have low efficacy towards other species that could be the reason for reproductive diseases such as *S. thoraltensis.*

Approximately 23% of all 56 *S. porcinus* isolates which were recovered in routine procedures showed a resistance to erythromycin by applying disk diffusion test in a Central European study [34]. This goes along with our results (17% resistant).

Although there is no current research on antimicrobial susceptibility of *S. orisratti*, we would expect a high similarity of resistance patterns between *S. orisratti* and *S. suis*. Nevertheless, resistance patterns between *S. suis* and *S. orisratti* were quite different in our investigations. However, we could recover only ten *S. orisratti* isolates.

There are no data on resistance rates of *S. alactolyticus*, which were tested by disk diffusion test. Belgian researchers demonstrated that detected porcine *S. alactolyticus* isolates carried less frequently tetracycline resistance genes than *S. dysgalactiae* and *S. suis* isolates [32]. However, in our investigations resistance rates of *S. suis* and SDSE to tetracyclines were slightly lower than those of *S. alactolyticus*.

## 4. Materials and Methods 

### 4.1. Study Design and Sample Collection

This is a retrospective study from data routinely collected by the University of Veterinary Medicine Vienna. These data include all 3141 samples from clinically affected pigs, which were bacteriologically examined at the Institute of Microbiology, University of Veterinary Medicine Vienna from January 2016 to July 2020. All samples were taken from clinically diseased pigs by 45 different veterinarians and were sent to the University of Veterinary Medicine Vienna for routine diagnostic purposes. Out of all 3141 examined samples, a total of 553 *Streptococcus* spp. were isolated (Table 4). Altogether, 419 isolates could be recovered from pigs that were necropsied either by the submitting veterinarian (*n* = 239) or at the University of Veterinary Medicine (*n* = 180). This includes all lung samples as well as samples from the central nervous system, joints, serosal surfaces and skin lesions. Another 52 isolates could be recovered from aborted fetuses, while a total of 82 isolates derived from samples taken from living animals directly on the farms, including swabs from noses (*n* = 46) and tonsils (*n* = 10) from pigs of all ages as well as swabs from the genitourinary tract of sows with reproductive disorders (*n* = 26). Blood samples and specimens from the liver, spleen, stomach, intestines, kidneys and heart were summarized as “other tissues”. *Streptococcus* spp. which were recovered from the intestines were not documented in routine diagnostic procedures, and are thus not presented in our results.

### 4.2. Species Identification and Antimicrobial Susceptibility Testing

Samples were incubated on blood agar BD Columbia Agar III with 5% Sheep Blood (BA) (Becton Dickinson, Heidelberg, Germany) under aerobic/microaerophilic (5% CO2)/anaerobic conditions, BD Chocolate Agar Columbia (Becton Dickinson, Heidelberg, Germany) under microarophil conditions at 37 °C for 24–48 h (sometimes up to 72 h), CNA Agar with 5% Sheep Blood, Improved II (Becton Dickinson), BD Mac Conkey II (MC) (Becton Dickinson) at 37 °C for 24–48 h and on Sabouraud dextrose agar with gentamicin and chloramphenicol (SAB (Becton Dickinson)) at 28 °C for 5 days. Samples were also incubated in Thioglycollate medium, enriched with vitamine K1 and haemin (Becton Dickinson).

Prior to susceptibility testing, presumptive *Streptococcus* spp. were identified to the species level by matrix-assisted laser desorption-ionization-time of flight mass spectrometry (MALDI-TOF MS) (Bruker Daltonik, Bremen, Germany). Species were included to the category “other streptococci” if the total number of detected isolates of the respective species was nine or less. This includes the species *S. henryi* (*n* = 5), *S. parauberis* (*n* = 5), *S. gallolyticus* (*n* = 4), *S. hyointestinalis* (*n* = 2), *S. pluranimalium* (*n* = 2), *S. salivarius* (*n* = 2), *S. uberis* (*n* = 2) *S. caballi* (*n* = 1), *S. gallinaceus* (*n* = 1), and *S. minor* (*n* = 1). Another isolate, which was only typable on a genus level, was also added to this group.

Susceptibility testing was performed for a total of 528 out of 553 isolates by agar disk diffusion according to the recommendations given by the CLSI documents M100 (26th ed., 27th and 28th ed.) (Clinical and Laboratory Standards Institute (CLSI), 2018). Human clinical break points (M100-28) were applied for the interpretation of zone diameters of eight tested antimicrobial substances. Antimicrobial substances were chosen due to given interpretative criteria for certain species for *Streptococcus* spp. and their importance to porcine health (penicillin, ceftiofur, tetracycline, erythromycin, levofloxacin and clindamycin) or to public health (linezolid and vancomycin) (Becton Dickinson, Heidelberg, Germany).

### 4.3. Evaluation and Statistical Analyses

All 553 *Streptococcus* spp. isolates, which were isolated in the course of routine diagnostic procedures from samples of clinically diseased pigs from January 2016 and July 2020, were selected for the study. All results were summarized retrospectively using TIS^®^ (Tierspitalinformationssystem Orbis VetWare, Agfa HealthCare, Bonn, Germany) and Microsoft Excel.

Descriptive data were assessed using SPSS (SPSS Statistics 25) [35]. Contingency tables were used to compare the frequency of the different *Streptococcus* species in the respective organs with the expected frequencies of an independent chi-square distribution. According to that, standard residuals (SR; cut off ± 1.96) were calculated to identify significant differences between counts and expected values in the individual combinations of pathogens and organs. Furthermore, Pearson’s chi squared tests were used to examine differences of resistances against tetracycline or erythromycin, depending on the species. Samples with intermediate resistance results were excluded from these analyses. In this context, *p*-values < 0.05 were considered to be significant. Contingency tables and Pearson’s chi squared tests were calculated in the R (Version 3.6.3) statistical computing environment [36].

## 5. Conclusions

Results show that different species of *Streptococcus* occur in different porcine organs and tissues. All isolates from skin lesions belonged to the species SDSE, which were also recovered significantly more often from specimens of the genitourinary tract than *S. suis*. *S. hyovaginalis* was most frequently recovered from the upper respiratory tract of weaned piglets and fattening pigs and *S. thoraltensis* from the genitourinary tract of sows. A clear statement if they are colonizers or potential pathogens cannot be given so far. However, especially the impact of SDSE and *S. thoraltensis* in the pathogenesis of reproductive disorders should be further investigated.

Due to observed differences in their resistance patterns, the identification of the *Streptococcus* species is highly recommended. Source of isolation might be also important since isolates from joints showed more resistances than others. High resistance rates of *S. suis* towards tetracyclines and erythromycin and high recovery rates of *S. suis* from lung tissue should be considered for the treatment of respiratory diseases in pigs.

## Figures and Tables

**Figure 1 antibiotics-09-00893-f001:**
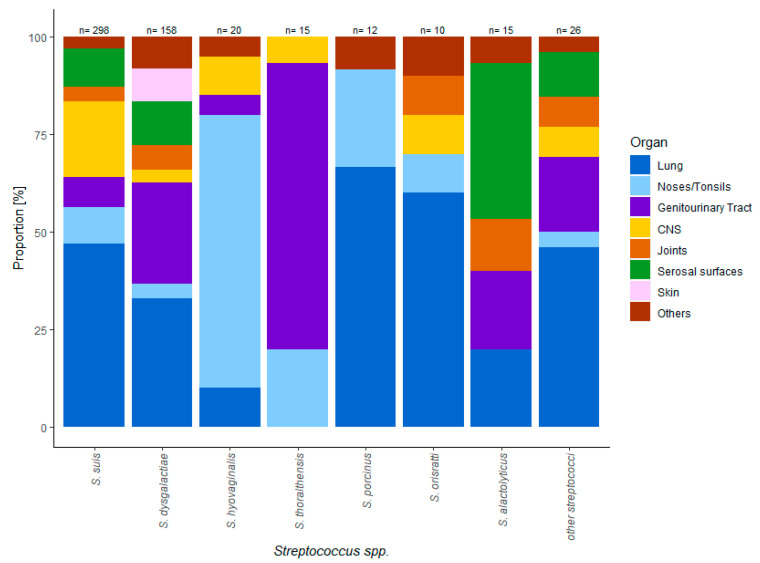
Organ distribution of different *Streptococcus* spp. isolates.

**Figure 2 antibiotics-09-00893-f002:**
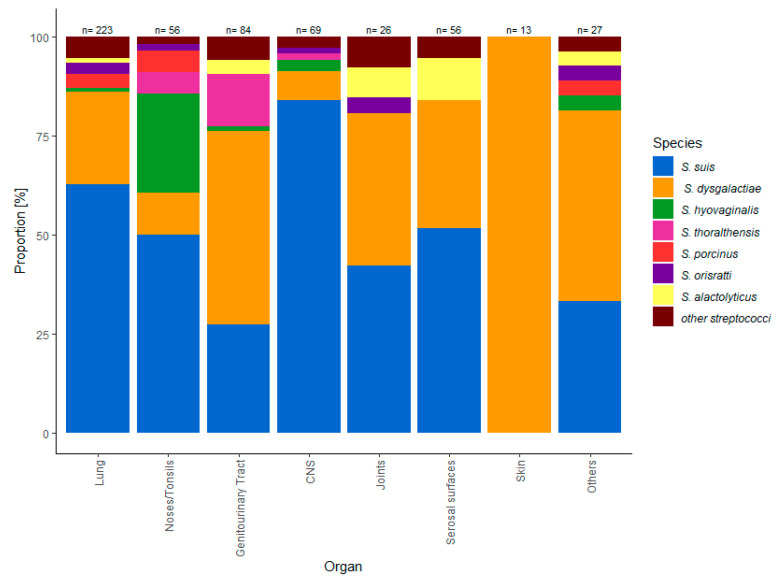
Distribution of the *Streptococcus* species in different organs.

**Table 1 antibiotics-09-00893-t001:** Total number of isolated *Streptococcus* species from different organs.

Organ	Tested (Total)	*S. suis*	SDSE	*S. hyovaginalis*	*S. thoraltensis*	*S. porcinus*	*S. orisratti*	*S. alactolyticus*	Other *S.* spp.
Lung	428	140	52	2		8	6	3	11
Noses/Tonsils	128	28	6	15	3	3	1		1
Genitourinary Tract	244	23	41		11			3	5
CNS	147	58	5	2	1		1		2
Joints	109	11	10				1	2	2
Serosal surfaces	217	28	18					6	4
Skin	49		13						
Others		9	13	1		1	1	1	1
Total		297	158	20	15	12	10	15	26

CNS—central nervous system.

**Table 2 antibiotics-09-00893-t002:** Frequency of resistances in the examined *Streptococcus* spp. [%] *.

Species	PEN	EFT	LVX	TET	ERY	CLI	N (Tested)
*S. suis*	1		1	66	58	62	297
SDSE			1	62	36	59	143
*S. hyovaginalis*				95	95	95	19
*S. thoraltensis*				91	55	73	11
*S. porcinus*				67	17	25	12
*S. orisratti*				100	88	88	8
*S. alactolyticus*	29	7		71	29	71	14
Others			9	75	50	57	24

PEN—Penicillin; EFT—Ceftiofur; LVX—Levofloxacin; TET—Tetracycline; ERY—Erythromycin; CLI—Clindamycin; * no isolate was resistant to Linezolid or Vancomycin.

**Table 3 antibiotics-09-00893-t003:** Frequency of resistant *S. suis* isolates in the relation to the tissue location [%] *.

Organ	PEN	LVX	TET	ERY	CLI	N (Tested)
Lung		2	65	52	58	140
Noses/Tonsils	7	4	75	79	89	28
Genitourinary Tract			78	48	43	23
CNS			53	57	53	58
Joints			82	73	64	11
Serosal Surfaces			57	50	57	28
Others			44	44	56	9

CNS—central nervous system; PEN—Penicillin; LVX—Levofloxacin; TET—Tetracycline; ERY—Erythromycin; CLI—Clindamycin; * no *S. suis* isolate was resistant to Ceftiofur, Linezolid or Vancomycin.

**Table 4 antibiotics-09-00893-t004:** Total number of examined organ samples from which *S.* spp. could be isolated and number of recovered *S*. spp. from every age group.

Organ	N (Examined Organs)	N (Total *S.* spp. Isolated)	N (Suckling Piglets)	N (Weaned Piglets)	N (Fattening Pigs)	N (Gilts/Sows)
Lung	428	223	6	154	59	4
Noses/Tonsils	128	56	1	21	19	15
Genitourinary Tract	244	83				83
CNS	147	69	12	54	1	2
Joints	109	26	5	17	3	1
Serosal Surfaces	217	56	6	40	7	3
Skin	49	13	6	6	1	
Other Tissues	1819	27	5	9	8	5

CNS—central nervous system.

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
