# Peer review of "Detection of Various Streptococcus spp. and Their Antimicrobial Resistance Patterns in Clinical Specimens from Austrian Swine Stocks"

_antibiotics, 2020, doi:10.3390/antibiotics9120893_

Round 1
Reviewer 1 Report
Manuscript well written with some minor changes
1) Concerning the references, please refer to the guidelines since the references are presented in different styles (ref 4, 8 20 for example) with abbreviations or not of the journal name ( with . or not), DOI or not.......
2)in Table 3 legend remove EFT definition since it does not appear in the table
Reviewer 2 Report
Renzhammer et al present a manuscript summarizing antimicrobial susceptibility patterns of Streptococcus species isolates from swine tissue or organ samples submitted to a veterinary diagnostic microbiology laboratory. This appears to be a descriptive observational retrospective study although the study design and sample sources are not adequately described.
Major concerns
Methods:
General – Reporting of observational studies should follow the Strengthening the reporting of observational studies in epidemiology – Veterinary Extension guidelines (STROBE-Vet, https://strobevet-statement.org/ ). Please add a STROBE-Vet reporting checklist and include information addressing the reporting recommendations in your manuscript.
Line 300 – please provide more detail on the source of the samples. Are these from necropsies submitted for diagnostic purposes? Are these samples from many different veterinary practices utilizing the diagnostic laboratory services? Is this simply a convenience sample? How were the isolates selected? Is this a retrospective study from data routinely collected by the laboratory, or is this a prospective study design addressing a specific hypothesis? Were all pigs in this study clinically diseased (this seems to be inferred at line 161 in discussion)? In what proportion of the cases was a primary bacterial infection identified? What was the age range of the pigs in the study? In some places you appear to be referring to tissues from neonatal pigs, and in others from adult pigs. Is there an association between the age category of the pig and the type of tissue examined? The methods need to provide more detail on the distribution of the source of the samples and thus the potential biases introduced by the sampling scheme. These sampling biases (due to differences in sample size and population) my explain some of the differences you observed compared to other studies (e.g. line 186). Other factors might include tupes of culture media or culture conditions used, but these are not reported in your manuscript so the reader cannot interpret your results in any context.
Line 310 - please find a reference for the use of MALDI-TOF MS to identify this range of streptococcus species isolated from swine, and provide more details on the interpretation of score values used for your study and thus the confidence in your species identification results. How many isolates received a score that indicated only a probable species level identification, or only a probable genus level identification? How many of these species were novel entries into the MALDI-TOF MS spectra database?
Line 313 – “Except for S. spp. isolated from lung tissue” – what break points were used for these exceptions? Which are approximately 40% of all isolates?
How were the drugs and classes selected in this study? Why was a cephalosporin not included (see your comment of ceftiofur use in swine production at line 242). Penicillin susceptibility is not a good proxy for cephalosporin susceptibility. Line 252 suggest you have data for ceftiofur, but these data do not seem to be included in the results.
Results
Figure 1 – what are “other streptococci”? are these isolates only typed to genus level by MALDI-TOF
Table 1 – the number of isolates sums to 552 – please confirm the values in the table to be consistent with txt at lines 16 and 71.
Table 2 the number of isolates sums to 528 – please indicate why 25 isolates were not included in this analysis
Presentation of susceptibility test results – in my opinion the actual values of antimicrobial susceptibility test results should be reported in summary tables. It is insufficient to report the proportion of isolates falling within categorical values (S,I, R) especially for organisms where there are no established breakpoints and breakpoints are based on extrapolations from human references. Please report the zone diameter distributions for each organism by drug tested. Examples of the format of tables commonly used for reporting such data can be found in Weinder et al doi: 10.1186/s13028-020-00533-3 (table 3) or various tables in Homer et al. doi: 10.1186/s12917-019-2162-8. This will require multiple tables, which could be added as supplemental data or as tables in the manuscript. Because breakpoints may change over time, reporting raw data summaries for susceptibility data is critical for any future comparisons with other studies. Further, by presenting the distribution of zone diameter values, the issue of having no interpretive criteria for some antimicrobials is avoided (e.g. trimethoprim-sulfamethoxazole as described at line 152) and all data from the study can be presented.
Data are incompletely presented - Table 4 lists 501 isolates associated with 7 tissues or organs, what tissues and organs are among the “others” for the 52 isolates not accounted for in table 4? But shown in figures 1 and 2 – please define the tissues or organs included among “others”
Minor issues
Line 16 – atypical – I do not understand the use of this term in this context – it seems inconsistent with descriptions in the introduction, which seems to suggest the species examined in this study are commonly isolated from pigs
Line 90 and 171 – grammar – Remarkable should be Remarkably, or better yet “We consider it remarkable that two…” or perhaps even delete – it is subjective whether something is remarkable or not? Just state the results.
Line 179 – “In a study of Brazilian colleagues” should be “In a study conducted by Brazilian colleagues”
Line 218 – delete “also”
line 243 – grammar
line 288 - grammar
References – genus species names are not written in italics in many references
Reviewer 3 Report
Thank you for the opportunity to review this study. The authors aimed to provide an overview of the antimicrobial resistance patterns and the distribution of various species of S.app. in different organs from clinically diseased pigs. This study will be helpful in understanding the distribution of S.app subspecies at different infection sites and in choosing appropriate treatment regimen. However, in order to better ensure the quality of the article, I have some minor problems as blow:
- In line 125-126, “Resistance rates of all other six investigated Streptococcus species towards these four antibiotics were also higher than to other substances”, There are four antibiotics mentioned here, but only three are listed above, please check it
- In materials and methods part, it is recommended to provide sampling location information and sampling methods. Sampling location information contributes to a better understanding of streptococcal infection epidemiology, and sampling methods determine the reliability of test results
- In line 333, the conclusion part, the conclusion statement does not reflect the results of the research. Further refinement and modification are recommended.
Round 2
Reviewer 2 Report
Thank you for more thoroughly addressing the epidemiology and provenance of the isolates.
I have no further comments or suggestions except i do fins two minor errors that might be addressed during final editting.
Line 66 – should be "there are few data"; data are plural
Line 388 – typographic error? some possible repetition in sentence